# Estimating Ensemble Flood Forecasts' Uncertainty: A Novel "Peak-Box" Approach for Detecting Multiple Peak-Flow Events

**Antonio Giordani [1,2,*]** , **Massimiliano Zappa [2]** and **Mathias W. Rotach [1]**

1   Department of Atmospheric and Cryospheric Sciences, Universität Innsbruck, 6020 Innsbruck, Austria; atmosphaere@uibk.ac.at
2   Swiss Federal Research Institute WSL, 8903 Birmensdorf, Switzerland; wslinfo@wsl.ch
*   Correspondence: antonio.giordani3@unibo.it

**Abstract:** Ensemble flood forecasts are an established tool to provide information about the uncertainty of runoff predictions. However, their interpretation may not be straightforward, especially when dealing with extreme events; therefore, the development of new tools to enhance their understanding and visualization is necessary. Recently, the so-called "peak-box" approach has been developed to help decision makers in the interpretation and verification of peak-flow forecasts, receiving positive feedbacks within the hydrological community. However, this method has proven to be limited when multiple peak-flow events occur within the forecast, being unable to separate close discharge peaks. The aim of this paper is then to develop a new algorithm designed to accomplish this task. To do so, we consider runoff probabilistic forecasts obtained with a coupled hydrometeorological flood forecasting system formed by the high resolution meteorological Ensemble model COSMO-E and the hydrological model PREVAH, for the small Verzasca basin, Switzerland, during October and November 2018. The application of this new method, despite the limitation given by the small sample size considered in this study, indicates a successful implementation: the new algorithm is able to distinguish among different events and to provide sharper and more skillful forecasts, and its verification yields slightly better timing estimations compared to the former approach.

**Keywords:** HEPS; flood forecast; peak-flow predictions; visual support; flood forecast uncertainty; model uncertainty; COSMO; PREVAH; peak-box approach

## 1. Introduction

The prediction of hazardous floods triggered by severe precipitation events is an important issue (e.g., [1]); especially for the Alpine region where the most severe events in Europe usually take place [2], the forecasting value has been increasing. Indeed, major floods can produce great damage both in terms of human and animal lives, as well as to the environment, economy, and infrastructures. Furthermore, one of the effects of the recent global warming is increasing the frequency of severe floods: Alfieri et al. (2015) [3] expected floods with current return periods longer than 100 years to double in frequency in Europe in the next three decades. In addition, exploring different European flood risk scenarios with a multi-model approach, Alfieri et al. (2018) [4] pointed out the accountability of climate change as the main driver influencing future flood risk and the expected increase of flood events' frequency even for the most optimistic warming scenario of 1.5 °C (compared to pre-industrial levels). Given the nature of the atmosphere as deterministic chaos [5], floods triggered by heavy precipitation events should be forecast with methods able to reflect this limitation in atmospheric predictability, and this is done through the employment of ensemble prediction systems (EPS or,

for hydrological purposes, HEPS). These models provide an ensemble of river flow predictions for the same forecast period, probabilistically assessing future river conditions [6], and are currently widely applied to obtain hydrological forecasts (e.g., [7–9]). There is general agreement that the probabilistic ensemble forecasting approach for flood warning purposes has added forecast value compared to the previously adopted deterministic forecasts (e.g., many case studies reported in [10]). Ensemble flood forecasts are affected by many sources of uncertainty [11,12], and the most important one is related to the numerical weather prediction (NWP) forecasts' input (especially regarding precipitation estimations) [13–16]. When comparing the weight of this uncertainty contribution with that of a minor source, related to the choice of different hydrological parameters (not shown here; the reader is referred to [17]), the importance of the meteorological input is highlighted. In fact, with the high-resolution forcing meteorological model we used, having an horizontal grid spacing of 2.2 km, the uncertainty related to the meteorological forcing was found to be five times larger than the hydrological parameters' one. Conversely, when using a coarser model with a 10 km horizontal grid spacing [13], the former uncertainty component was found to be ten times larger than the latter. This result is mainly imputable to a finer horizontal grid spacing of the forcing meteorological model, which leads the precipitation predictions to outperform those of the coarser one significantly (e.g., [18,19]). Anyway, despite these significant improvements, ensemble flood forecasts are still usually characterized by large spread, especially in the case of extreme events; consequently, their interpretation may not be so straightforward. A tool that has been developed to enhance the understanding of probabilistic runoff forecasts is the "peak-box" approach [20], especially for decision-making and to guide forecasters in the interpretation of ensemble peak-flow forecasts. Based on the peak magnitude and timing distributions among the ensemble realizations, this visual tool has proven to be useful for flood peak estimation [21,22], but still, it is limited for events characterized by multiple flood peaks. In fact, it is plausible to observe multiple and close discharge peaks caused by intermittent storms (e.g., [23,24]), and the peak-box is not able to separate them automatically in order to produce peak estimations related to different events. For this reason, the open questions we want to address in this paper are:

> Is it possible to develop a peak-box approach for detecting multiple flood peaks within the same runoff ensemble forecast, and does it actually outperform the former method of Zappa et al. (2013) [20]?

To achieve this goal, we adopted a flood forecasting system formed by COSMO-E (i.e., the Ensemble version of COSMO model) as the forcing NWP and the hydrological model PREVAH to produce runoff simulations on a small Alpine river catchment. These forecasts were used to develop and test a more sophisticated peak-box algorithm, with the scope to distinguish and provide interpretations of runoff events characterized by multiple peaks.

The manuscript is organized as follows: in Section 2, the methods, comprising the new peak-box algorithm, are described; in Section 3, the results of the new method and a comparison with the former approach are reported; in Section 4, the results are discussed; and in Section 5, the conclusions are drawn. In addition, in Appendix A, an assessment of the quality of the forecasts considered is reported.

## 2. Methods

### 2.1. Flood Prediction Chain

The flood forecasting system considered for this study is formed by:

- COSMO-E (COnsortium for Small-scale MOdeling) as the forcing NWP system;
- PREVAH (precipitation-runoff-evapotranspiration HRU (i.e., hydrological response unit) model) as the hydrological model.

COSMO-E is the operational high resolution limited area ensemble prediction system of, e.g., MeteoSwiss (e.g., [19]). It uses the non-hydrostatic COSMO-2 model version (e.g., [18]), with a

2.2 km horizontal grid spacing, has 21 ensemble members, and is run over the entire Alpine region twice daily (at 00 and 12 UTC) up to a lead time of 120 h. The initial and boundary conditions of COSMO-E are downscaled from the 51 member global ensemble model ECMWF EPS (e.g., [25]). Especially important for our tasks is the precipitation forecast produced with hourly steps by the NWP model, which is then spatially interpolated by the hydrological model over the catchment considered: the spatially distributed meteorological data are averaged on defined sub-areas depending on different altitude zones, which, for small basins as the one considered here, are identified with 100 m elevation bands [26]. For a detailed description of the COSMO model and the physical parameterizations adopted, the reader is referred to the last updated version of [27].

PREVAH is a semi-distributed hydrological catchment modeling system developed especially to run simulations in mountainous environments. To generalize the local runoff generation behavior over the entire basin considered, the 500 m$^2$ grid points of PREVAH are aggregated to HRUs (which represent a division of the basin into areas presenting similar hydrological behaviors). The reader is referred to [28] for a detailed description of the model, its physics and parameterizations, and previous work reviews. The initial setup and calibration of PREVAH for the study area considered relied on previous works [29,30]. The calibration considered was thought to obtain more accurate predictions during the peak phases and to give less importance to the baseflow phases (i.e., the so-called "flood calibration", based on peak-flow sensitive efficiency scores, and described in detail in [31,32]). For PREVAH, a set of sensitive parameters was selected and randomly perturbed through a Monte Carlo experiment to obtain 25 different sets of parameters as in [13]. The 25 hydrological parameter sets we used represent the 1% subset of Monte Carlo realizations that performed best during the calibration period 1996–2001 on the studied catchment.

After a complete run of the flood forecasting system, we obtained a number of realizations given by the product of the ensemble members of COSMO-E and the number of parameter sets of PREVAH, i.e., 525 runoff simulations, which had a lead time of 120 h.

### 2.2. Study Area and Period

The study area was the Verzasca basin (Figure 1), located in the southern Alps in the Canton Ticino, Switzerland, which covers an area of 186 km$^2$ and spans an elevation range between 490 and 2900 m a.s.l. This catchment is relatively little influenced by human activities, and the land use consists of 30% forest, 25% shrub, 20% rocks, and 20% alpine pastures [29]. The choice of this basin was due to its character of being relatively prone to flash floods. Its discharge regime consists mainly of snowmelt during the hot season (spring and early summer) and of large rainfall events in autumn.

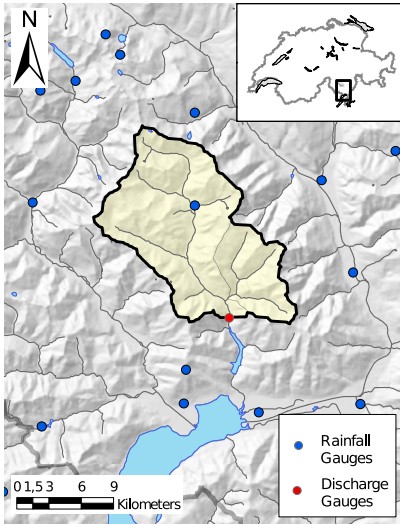

**Figure 1.** The Verzasca basin (in yellow). The runoff gauge, where discharge observations were detected, is indicated with a red solid dot.

The flood forecasting system was run from 2018-10-23 12:00 UTC to 2018-11-28 12:00 UTC, with the exception of 2018-11-01 00:00 UTC (for technical reasons), obtaining a series of 72 different model initializations (twice daily at 00 and 12 UTC), referred to as the full cases (which were considered to assess the quality of the forecasts in Appendix A). However, for the purpose of the peak-box application carried out, the subset of 35 initializations showing sufficiently large values of discharge to serve our purpose, spanning from 2018-10-23 12:00 UTC to 2018-11-10 12:00 UTC, was selected and in the following referred to as the high-flow cases. This period was particularly interesting because at the end of October and beginning of November 2018, the Alpine region was affected by extreme rainfall and flooding events caused by an intense cyclone formed over the Mediterranean sea (see the EUMETSAT web page (https://www.eumetsat.int/website/home/Images/ImageLibrary/DAT_4169486.html) for a detailed description).

To distinguish between low flow conditions and runoff events, we used the threshold $T_{low} = 23.1$ $m^3 s^{-1}$, which is the mean discharge value of the Verzasca basin, i.e., climatological value over the period 1990–2016, which has been exceeded on 10% of the days within a year.

### 2.3. Observed Data

Runoff observations were employed to compare with forecasts output and to verify simulation results. They were obtained from the gauging station present in Lavertezzo, Campiòi (red dot in Figure 1), maintained by the Swiss Federal Office for Environment and supplying measurements at 10 min resolution, which were then hourly averaged.

### 2.4. Quality of the Runoff Predictions

The quality of the ensemble runoff forecasts is investigated in some detail in Appendix A; here, we report the main findings. The main limitation of the analysis carried out was given by the small sample size of model initializations considered, which restricted the general validity of the results. The discharge predictions produced with the coupling of COSMO-E with PREVAH were revealed to be highly skillful in detecting runoff events or rejecting non-events. There was, however, a general tendency to systematically underestimate runoff's magnitude in the specific Verzasca basin, due to a miscalibration of the hydrological model in the catchment.

### 2.5. The Peak-Box Approach

Forecasts of flood events are usually characterized by a large ensemble spread; consequently, their evaluation may be difficult. Therefore, a visual procedure to evaluate ensemble flood peak and flood timing was developed by Zappa et al. (2013) [20], under the name of "peak-box", with the purpose of helping decision-making when forecasting flood events. In the following, we will refer to the procedure developed in the paper as the "classic" (PBC) benchmark, against which the newly developed peak-box algorithm for detecting multiple flood peaks (PBM) was tested. Firstly, we describe the structure of PBC as developed in [20], followed by the description of PBM. For both approaches, the threshold $T_{low}$ was adopted to discard the predicted peaks relative to low flow conditions and to keep those producing useful information for flood forecasting purposes.

#### 2.5.1. The Classic Peak-Box

When a decision-maker faces any flood forecast, the crucial question they should be able to answer is: "How high and when is the flood expected?". In fact, peak discharge $p$ and peak timing $t$ are the most important information to have for planning flood relief measures [20]. Considering ensemble forecasts, to answer this question, one has to take into account the entire distribution and relative spread of the ensemble forecasts produced. This is the main reason behind the development of the peak-box approach, which aims at visualizing forecast uncertainty related to peak-flow and which is sketched in Figure 2, left panel. When plotting all ensemble generated discharge forecasts as a function of time, the "peak-box representation" adds four elements:

1. the outer rectangle, called the "peak-box", having the lower left coordinate set to $(t_0, p_0)$, i.e., the earliest time of peak-flow occurrence in any of the ensemble members, $t_0$, the lowest peak discharge, $p_0$, and the upper right coordinate set to $(t_{100}, p_{100})$, i.e., conversely, the latest time of peak-flow occurrence, $t_{100}$, and the highest peak discharge, $p_{100}$, among all the ensemble members and for the entire forecast period;

2. the inner rectangle, the IQR box (i.e., interquartile range box), which has the lower left coordinate set to $(t_{25}, p_{25})$, i.e., the 25% quartile of peak timing, $t_{25}$, and discharge, $p_{25}$, and the upper right coordinate set to $(t_{75}, p_{75})$, i.e., the 75% quartile of peak timing, $t_{75}$, and discharge, $p_{75}$, among all the ensemble members and for the entire forecast period;

3. the horizontal line, ranging from $t_0$ to $t_{100}$, representing the median of the peak discharge ($p_{50}$) of all members of the ensemble forecast;

4. the vertical line, ranging from $p_0$ to $p_{100}$, representing the median of the peak timing ($t_{50}$) of all members of the ensemble forecast.

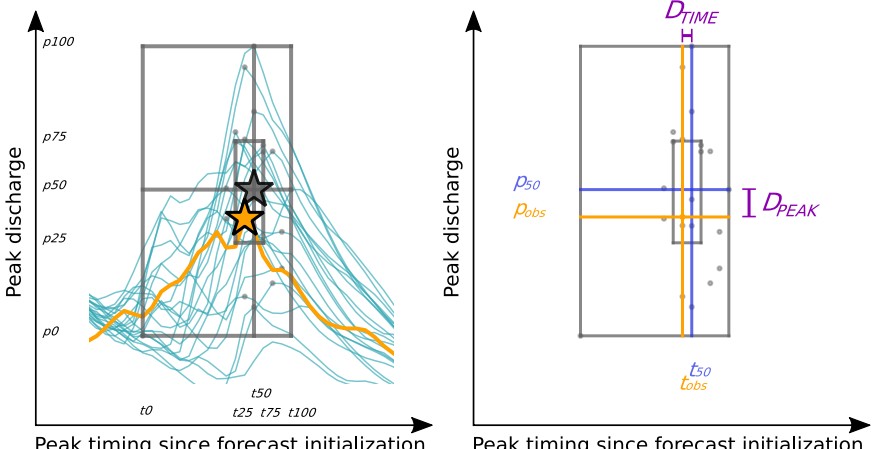

**Figure 2.** Application of PBC to an ensemble of runoff simulations. Left panel: implementation of the peak-box (grey solid lines and star) to a runoff forecast characterized by many members (sea-green solid lines), for which the observed runoff is also reported (orange line), together with the predicted peaks (small grey dots) and the observed peak (orange star). $t_{xx}$ and $p_{xx}$ define the 0%, 25%, 50%, 75%, and 100% quartiles of the peak-flows, both on the *x*-axis, representing the time since forecast initialization, and the *y*-axis, representing the peak discharge. Right panel: estimation of measures of agreement from the peak-box. The geometric meaning of the two measures of agreement $D_{TIME}$ and $D_{PEAK}$ is illustrated. See the text for further details. Adapted from [20].

Together with the peak-box, some metrics to quantify the forecast sharpness and to verify its prediction were defined and are sketched in Figure 2, right panel. Since sharp forecasts increase the confidence in decision-making compared to ensembles having a larger spread, it makes sense to measure the peak-flow forecast sharpness through the results of the peak-box approach: considering the ranges of peak timing and discharge obtained, we can measure peak-box and IQR-box sharpnesses as:

$$PB_{FULL} = (p_{100} - p_0) \cdot (t_{100} - t_0) \frac{3.6}{A} \quad [\text{mm}] \tag{1}$$

$$PB_{IQR} = (p_{75} - p_{25}) \cdot (t_{75} - t_{25}) \frac{3.6}{A} \quad [\text{mm}] \tag{2}$$

where the factor $3.6/A$ is a scaling parameter ($A$: catchment area in km$^2$) used to obtain millimeters of water depth and to compare results from different basins. Ensemble peak-flow estimations can also be verified through the usage of the peak-box. One verification metric was introduced simply by checking whether the observed peak fell inside or outside the boxes (i.e., adaptation of other well known categorical verification metrics (e.g., [33])): if the observed peak fell outside the peak-box, it was

labeled with a "miss", if it fell inside the peak-box with a "hit", and if even inside the IQR-box with an "IQR-hit". As another metric, we considered the peak ensemble median ($t_{50}$, $p_{50}$) as the best guess of the forecast for predicting the true runoff peak ($t_{obs}$, $p_{obs}$): the definition of the scores in Equations (3) and (4) permits estimating the level of agreement between the observation and the ensemble median and, consequently, the accuracy of peak timing and discharge predictions.

$$D_{PEAK} = |p_{50} - p_{obs}| \quad [\text{m}^3\text{s}^{-1}] \tag{3}$$

$$D_{TIME} = |t_{50} - t_{obs}| \quad [\text{h}] \tag{4}$$

### 2.5.2. A New Algorithm for Multiple Peak-Flow Events

A critical limitation of PBC is its inability to interpret peak-flows when multiple peaks are predicted from the same ensemble simulation. This was the reason that prompted us to develop a more sophisticated algorithm able to provide reliable estimations for these kinds of situations. The procedure, based on peaks' detection and separation into groups related to different runoff events, involved the application of the following steps, sketched in Figure 3:

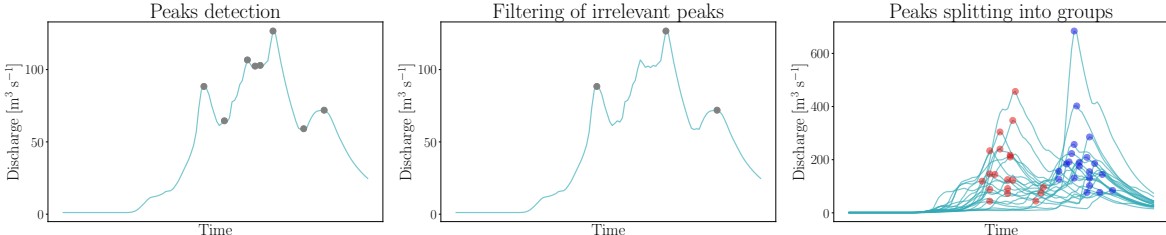

**Figure 3.** The steps of PBM for forecasting multiple peak-flow events for a specific forecast initialization. The grey dots represent the peaks detected for the realization considered. Left panel: peaks detected without the application of any filtering. Center panel: peaks detected with the condition of a certain peak prominence, i.e., "important" peaks. Right panel: peaks detected and split into groups related to different events (represented by different colors) for the entire pool of ensemble members, comprising also the realization reported in the first two panels (notice the different discharge range covered by the *y*-axis).

1. For every ensemble member, find all the runoff peaks (i.e., local maxima), excluding the first and last hours of the forecast. The peaks were selected based on the concept of peak's topographic prominence: the prominence, defined as the minimum height necessary to descend to get from the summit to any higher level terrain [34], is a measure of the independence of a peak. Its application, when detecting the local maxima of a curve, permitted filtering out irrelevant and noisy peaks. The higher the prominence, the more "important" the peak is considered. For our purposes, within Python's SciPy function `scipy.signal.find_peaks` (https://docs.scipy.org/doc/scipy/reference/generated/scipy.signal.find_peaks.html), we set a value of prominence of 1 for the first and last five hours of the forecast (since at the temporal extremes, some peaks may be discarded when considering too large prominence values), a value of 2 in the case of ensemble members whose discharge's absolute maximum was below 100 m$^3$ s$^{-1}$, and a value 8 for all the rest. To visualize the effect of prominence on peak detection, we can compare the left and the center panels of Figure 3: while on the left panel, the peaks found comprise many negligible peaks, on the center panel, only the relevant peaks are kept through the application of the aforementioned procedure.

2. Since the peak-box is thought to forecast high-flow events, to avoid low-flow conditions, the threshold $T_{low}$ to reject exceptionally low peaks was applied to every realization. Furthermore, if in the vicinity of a peak (i.e., in the temporal window of $\pm 10$ h around it), other peaks were present, only the peak presenting the highest discharge would be kept.

3. Considering all the runoff realizations together, apply a K-means clustering [33,35] to the peaks' population to separate them into groups related to different peak-flow events (Figure 3, right panel). The variables on which the clustering was performed were the peak time of occurrence and the scaled peak runoff. Scaling the peak discharge by a factor of 10 was applied in order to enhance the weight of the time variable [36]. This was chosen because for the clustering we aimed at, the scope was to split the peaks into groups related to events close, but distinct in time. Since for a K-means clustering, the number of clusters into which we split the data must be chosen in advance, we prescribed the number of groups into which the forecast peaks were divided by extracting the rounded mean value of the number of peaks among all the ensemble members. In a group, we allowed only one peak for each ensemble member: if more than one was present, only the peak having the largest discharge value would be kept.

4. For every group found, apply the PBC procedure to construct the boxes and to calculate the measures of sharpness. Concerning peak-flow verification, an additional condition was prescribed due to the increased multiplicity of the observed peaks (which were identified with the application of Steps 1 and 2 to the observed runoff time series): if more than one observation fell inside a peak-box, the verification of the estimated peak was performed against the closest, both in time and in runoff magnitude, observed peak. This condition was applied to both PBC and PBM forecast verifications.

We are aware that this last condition was not a proper objective criterion. However, with the scope of the present paper being to check whether PBM shows significant improvement compared to PBC and since with PBC, only the highest observation was considered, verifying the prediction against only the nearest observation was justified in this case. For any subsequent purposes beyond this work, this limitation shall be accounted for. Both PBC and PBM were applied to the high-flow cases' set of model initializations. Given the increased multiplicity of peaks obtained with PBM, both in terms of observations and forecasts, it was clear that the hit/miss score developed for PBC could not give reliable results in this case. For this reason a new score, referred to in the following as $H_{PB}$, was defined for both the methods as the number of hit peaks (i.e., observed peaks falling inside a peak-box) divided by the total number of observed peaks within a forecast:

$$H_{PB} = \frac{\text{number of hit peaks}}{\text{number of observed peaks}} \qquad (5)$$

The Python script developed following this procedure is freely available at the GitHub repository: https://github.com/agiord/peakbox.

The application of both the approaches was carried out with the runoff ensemble forecast comprising just the 21 meteorological medians out of the 525 model predictions. The meteorological medians were calculated from those realizations sharing the same COSMO-E member, but employing different hydrological parameters sets. This was done firstly to save computational resources that such a larger ensemble would demand, but also since the effect of the hydrological uncertainty was only to change the magnitude of the predicted discharge peaks slightly, leaving their timings unperturbed (at least for the cases treated here). For this reason, we assumed that applying the method to the reduced ensemble instead of the full ensemble did not lead to a significant loss of information.

## 3. Results

In this section, the results pertaining to the application of the newly developed peak-box approach PBM for detecting multiple peak-flows and a comparison with the classic approach PBC are reported.

The results of the application of the peak-box approach to four different model initialization times are reported in Figure 4. Panels (a), (b), and (c) pertain to the forecast of the main runoff event in the period considered. This event was particularly relevant since it was characterized by two main discharge phases, separated by a short period with lower runoff (approximately around the night of 29

October); consequently, it perfectly served to test the new method. Panel (d) pertains to the forecast of a minor isolated event. For each initialization, PBC produced always one peak-box (by definition), which should predict the highest observed peak within the forecast. However, this was not always the case: e.g., in Panel (c), PBC's estimation is found to be closer, both in terms of timing and magnitude, to the second highest observed peak. Furthermore, the presence of many observed peaks within the same forecast led to the production of wide PBC boxes. This is the case of Panel (b), where the box resulting from the distribution of the peaks related to four different observations spans almost four days of the forecast, thus producing a peak estimation, totally missing sharpness. PBC was confirmed to produce reliable and sharp forecasts in the case of singular runoff events, as is the case in Panel (d).

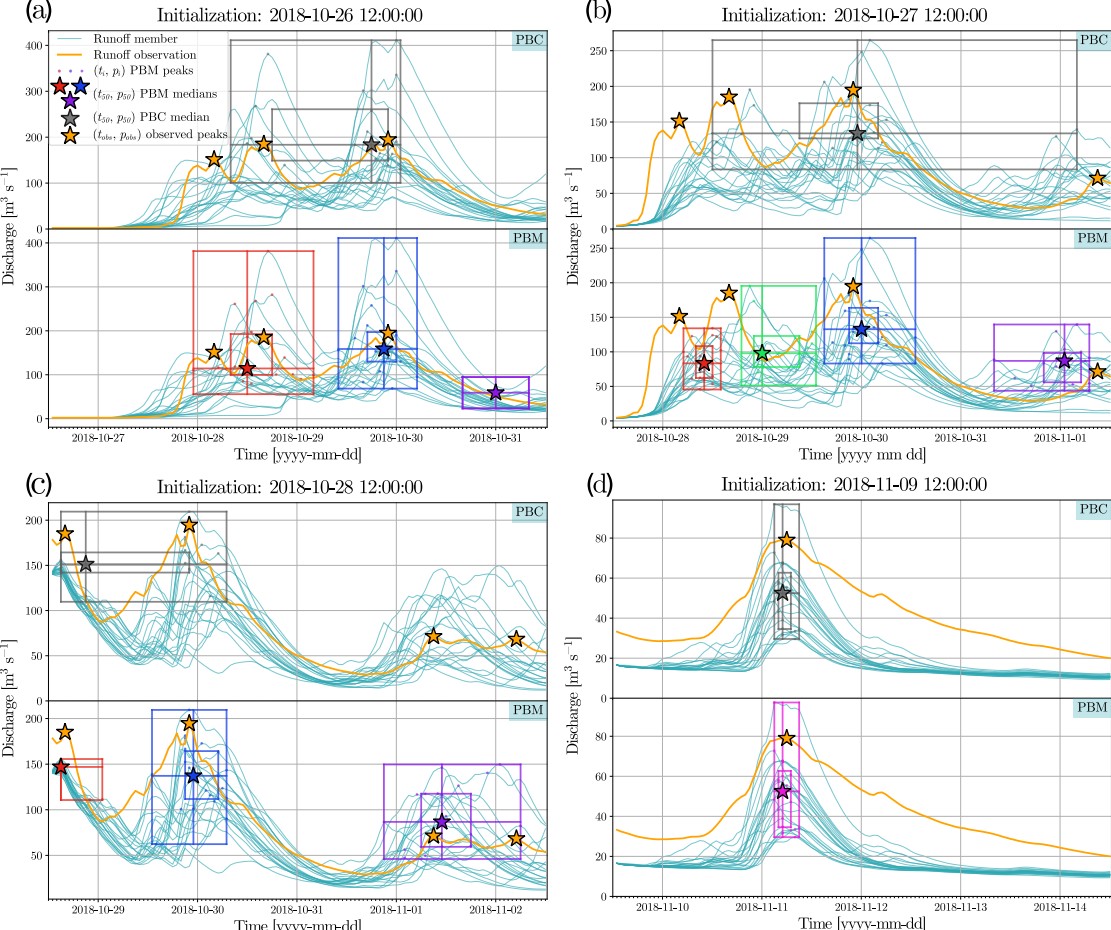

**Figure 4.** The application of the peak-box approach to a set of four model initializations. For every model initialization (**a**–**d**), the resulting ensemble flood forecast, pertaining only to the 21 meteorological medians, is shown with thin sea-green solid lines; the observed discharge is indicated with an orange solid line; while the detected peaks ($t_i$, $p_i$), from which relative peak-boxes are constructed, are reported with small solid dots. The plots are divided into two parts: the upper panels pertain to the application of PBC, in grey solid lines, while the lower panels of PBM are in multiple colored solid lines. Every peak-box forecast contains the peak-box (outer rectangle), the IQR-box (inner rectangle), and the median estimations of peak timing and discharge (vertical and horizontal solid lines). The crossing between these two lines in ($t_{50}$, $p_{50}$) represents the most probable peak estimation for every peak-box and is highlighted with a star. The orange stars represent the observed peaks (detected with the same procedure used for the detection of predicted peaks). The legend reported in panel (**a**) extends also to the other panels.

Focusing now on PBM forecasts, the new algorithm always produced more than one box for forecasts characterized by multiple runoff events (Panels (a), (b), and (c)). The generally sharper

peak-boxes produced with PBM successfully distinguished among the main events, if they were not excessively too close in time (as is the case of the first two observed peaks in Panel (a) or the last two in Panel (c)). The ability of the peak-boxes to hit the observed events depended on the spread of the ensemble forecast: e.g., in Panels (b) and (c), at the beginning of the forecasts, the spread is too small to produce a correct peak estimation for the first event, while, at later lead times characterized by sufficient spread, PBM produces skillful peak-boxes (i.e., hitting the observations) more easily. However, for the former cases, it was relevant that PBM was able to recognize the presence of peak events, producing their corresponding (but unskillful) peak-boxes. Finally, PBM was also able to reduce correctly to PBC when just one runoff event was present within the forecast (Panel (d)): the peak-box forecast produced with PBM matched perfectly the one obtained with PBC, both in terms of the sharpness of the box and peak median.

### 3.1. Forecast Sharpness

A comprehensive comparison between boxes' sharpness of the two approaches including all the high-flow cases is offered in Figure 5. Overall, PBM tended to produce sharper forecasts than PBC, both in terms of $PB_{FULL}$ and $PB_{IQR}$. The mean values of both sharpness metrics were almost three times smaller than the mean values related to the PBC boxes. Furthermore, almost for all the cases where PBM created just one box, the values of peak-box sharpness of the two methods resulted in being equal, representing the correct reduction of PBM to PBC. Generally, the sharpness of PBM forecasts did not seem to be influenced by forecast lead time: as can be seen by the dot size distribution in Figure 5, many times, the sharpest peak forecast within the same initialization is not related to the first occurring peak-box.

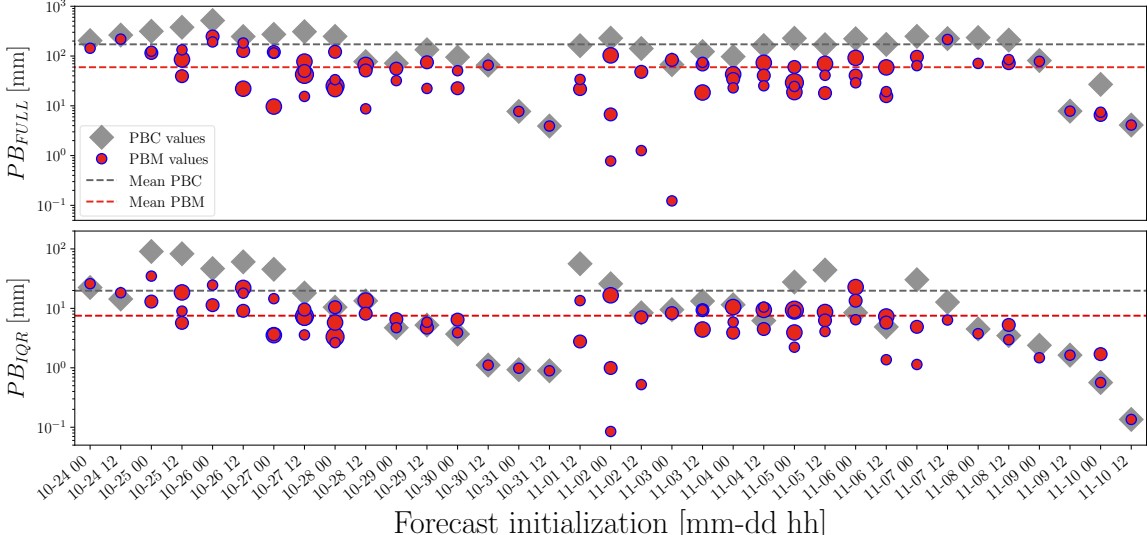

**Figure 5.** Measures of sharpness for the peak-box, $PB_{FULL}$ (upper panel), and the IQR-box, $PB_{IQR}$ (lower panel), for both PBC (grey diamonds) and PBM (red dots), for every forecast initialization of the high-flow cases. The size of the PBM dots refers to different lead times within the same forecast initialization: the larger the size of the dot, the later the lead time when the peak-box is placed. The mean values of $PB_{FULL}$ and $PB_{IQR}$ are shown with horizontal dashed lines, grey for PBC and red for PBM.

### 3.2. Events' Detection

The $H_{PB}$ values reported in Figure 6 reveal PBM to perform always better or equal to PBC in detecting the observed peaks, resulting in mean $H_{PB}$ values of 0.52 for PBM and 0.41 for PBC. This means that PBM was able to detect half of the observed peaks successfully, while PBC just 40%. Concerning IQR-box hits (not shown in the chart), PBM successfully detected 20% of the observed

peaks, while PBC only 17%. On the other hand, PBM produced also a larger amount of misses than PBC (empty boxes in Figure 6). PBM outperformed PBC, in terms of $H_{PB}$, especially in about the first half of the high-flow cases, which were the forecasts capturing the highest runoff events and characterized by multiple peak-flows (as in Figure 4, Panels (a), (b), and (c)). During about the second half of the forecasts, PBM performed equally to PBC in many cases, with the observed peaks being lower in magnitude and their amount being decreased, if not even reduced to just one peak per forecast (as the case of Figure 4, Panel (d)).

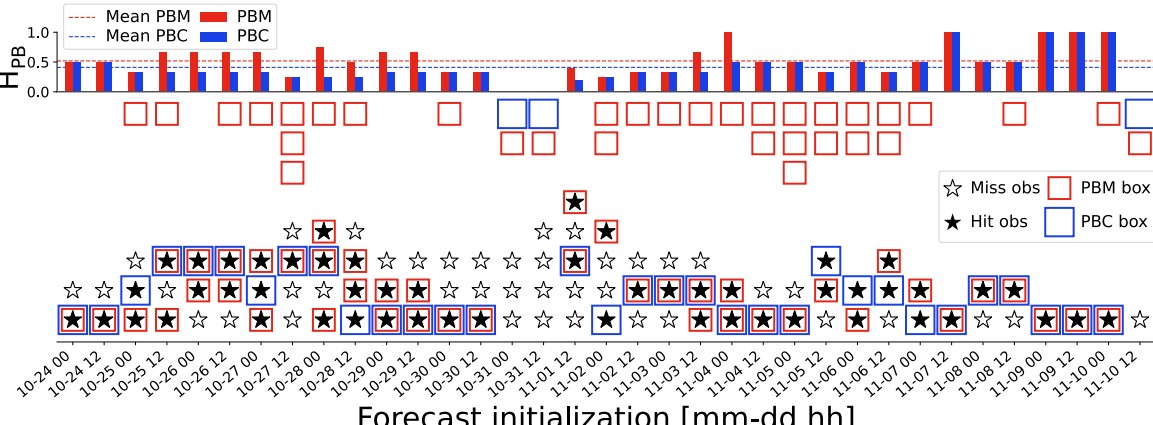

**Figure 6.** Hit and miss observed peaks on the high-flow cases peak-boxes. On the upper part of the plot, the hit score $H_{PB}$ is shown as a bar plot both for PBC (in blue) and PBM (in red), together with the resulting mean values (dashed horizontal lines). On the lower part of the plot, for every initialization, all the missed and hit observations are reported (white and black stars, respectively), which are ordered from bottom to top with increasing lead time of occurrence. Together, the corresponding approach that detected them is shown (red boxes for PBM, blue boxes for PBC), as well as the amount of peak boxes that did not detect any event (empty red and blue boxes in the upper part of the lower plot). The cases in which both blue and red empty boxes are obtained correspond to the occasions in which PBM reduces to PBC, and both miss the observation, so the empty boxes of the two methods result in being exactly equal and occurring at the same lead time.

### 3.3. Peak Median Verification

The median in timing and peak magnitude was considered, within the predicted peaks of the same box, as the best estimation of an event, and it was the object of the verification metrics $D_{PEAK}$ and $D_{TIME}$. The outcomes of these metrics, both for PBC and PBM, are summarized in Figure 7. The verification values reported, both for discharge and timing, included just the skillful cases (i.e., when an observed peak was detected inside a peak-box). Concerning $D_{PEAK}$, we see that on average, the performance of PBM was very similar to PBC (i.e., the horizontal dashed lines on the upper panel are superposed). Concerning $D_{TIME}$, PBM predicted on average peak timings closer to the observed peaks of slightly less than two hours with respect to PBC. In four cases, PBM predicted perfect peak timing estimations (i.e., $D_{PEAK} = 0$ h). Similarly to what was obtained for the sharpness metrics, also for the verification metrics, there was no significant dependence of the forecast quality with lead time. In fact, the dot size distributions depicted in Figure 7 reveal that the peak-box estimation producing the lowest $D_{PEAK}$ and $D_{TIME}$, in those cases with more than one peak-box, is often not the first produced within the same forecast. The comparison between PBM and PBC for predicting better or equal timing and/or peak estimations, considering all the boxes produced by PBM and at least one box produced by PBM, is shown in Table 1. PBM performed better than PBC in producing at least one skillful box for 66% of the times. Concerning the estimations of at least one skillful value of discharge and timing within the formed boxes, PBM performed better than PBC respectively in 69% and 81% of the cases.

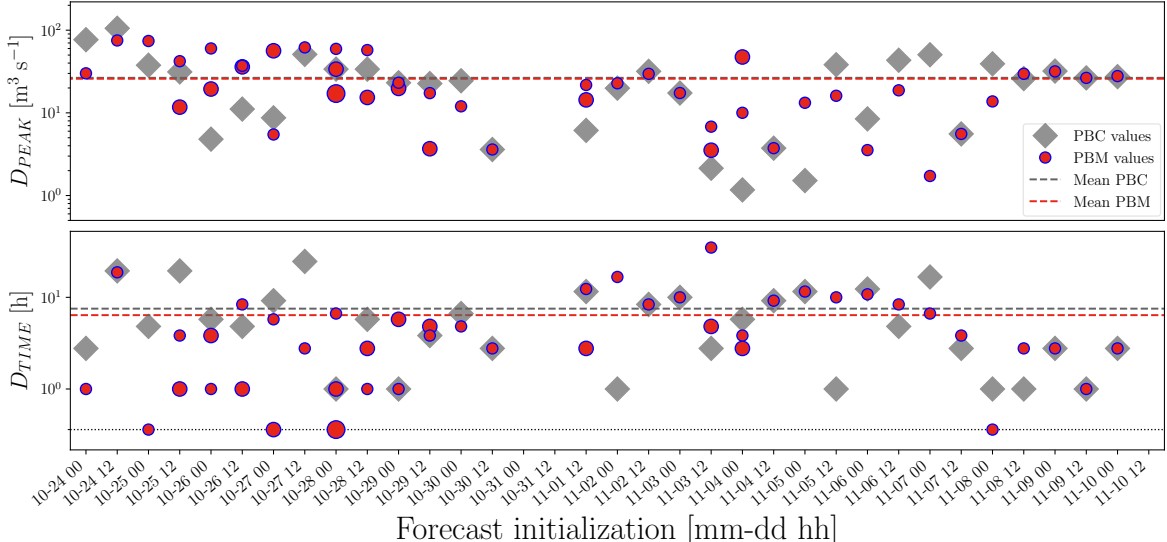

**Figure 7.** As in Figure 5, but for measures of peak verification $D_{PEAK}$ (upper panel) and timing verification $D_{TIME}$ (lower panel). The black thin dotted horizontal line in the lower panel indicates the value $D_{TIME} = 0$ h.

**Table 1.** Amount of PBM forecasts performing better than or equal to PBC (in terms of $D_{PEAK}$ and $D_{TIME}$) within the group of skillful forecasts of the high-flow cases. They are reported considering the performance of all PBM boxes together (upper row) and of at least one PBM box (lower row) per forecast, separately for peak timing and discharge, as well as for both peak metrics together.

|  | $D_{PEAK}$ | $D_{TIME}$ | $D_{PEAK}$ and $D_{TIME}$ |
|---|---|---|---|
| PBM $\geq$ PBC for all the boxes | 56% | 66% | 38% |
| PBM $\geq$ PBC for at least 1 box | 69% | 81% | 66% |

## 4. Discussion

In the following section, the results found concerning the assessment of the new PBM algorithm are discussed.

### 4.1. Forecast Sharpness and Peak Median Verification

Overall, PBM produced sharper forecasts than PBC, mainly because PBC considered the distribution of just the absolute maxima among the ensemble forecasts to produce a peak-box, while PBM took into account the entire distribution of the local maxima among the different runoff members. Consequently, when PBM split all the peaks found in different temporal clusters, the spread of a single peak-box produced through PBM resulted in being significantly smaller than PBC's outcome, especially concerning peak timing uncertainty. Concerning the verification of the predicted peak medians, the lack of enhancement of PBM's peak magnitude estimations compared to PBC was imputable to the model itself. In fact, the forecasts showed a general tendency to underforecast river runoff in the Verzasca basin, thus leading to an underestimation of the peak magnitude of about 26 $m^3 \, s^{-1}$ when compared to the observed events. This was demonstrated in previous studies, where a generally better peak timing estimation and an underestimation of peak discharge were found when applying PBC for the same modeling chain we used on the Verzasca basin [37]. The independence found in PBM's peak forecast sharpness and median verification from lead time was in contrast to what was found by the authors formulating the PBC approach. Zappa et al. (2013) [20], in fact, detected a significant drop in forecast sharpness for longer lead times when applying PBC to the Verzasca basin. The reason for this may be found in the enhancement of the grid spacing of the forcing meteorological model adopted: while in [20], the meteorological driving model was COSMO-LEPS (i.e., COSMO

Limited area EPS), we used the much better resolved COSMO-E, which was found to be superior to its predecessor in predicting precipitation events [37]. Unfortunately, a direct comparison between the resulting lead time evolving spreads of the two models could not be made, so we hypothesized that COSMO-LEPS's resulting spread was significantly larger for the longest lead times, causing the drop in PBC's sharpness and median verification quality detected.

### 4.2. Events' Detection

Concerning the ability for events' detection, PBM resulted in being a more skillful method than PBC and produced more than one peak hit within the same forecast when needed, even if also a larger amount of unskillful boxes was produced by PBM. Furthermore, the smaller amount of misses obtained with PBC was imputable to the lack of sharpness of this method, and not to its actual ability to produce less unskillful and more reliable forecasts. In fact, it was clear that in the extreme case of a peak-box extending for the entire 120 h of the forecast (i.e., a forecast totally missing sharpness), the highest event would always be captured. The general inability to detect events taking place in the first 24 to 48 h of the forecast successfully was due to the smaller amount of spread presented in the short range by COSMO-E's precipitation forecasts [19]. In fact, this NWP forcing model's feature remained also after the propagation through PREVAH, as can be seen from Figure 8: during the first phase of the forecast, the spread resulted in being much smaller than the median runoff value, then it progressively increased, reaching the maximum around Day 4 of the forecast, and then, it intermittently decreased during the last hours of the prediction.

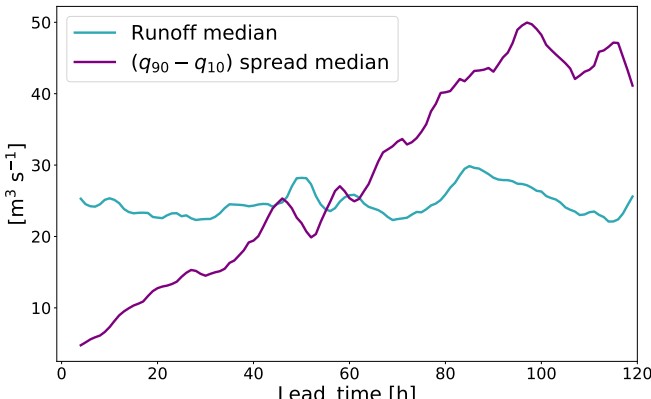

**Figure 8.** Runoff median (sea-green line) and ($q_{90} - q_{10}$) spread median (purple line) as a function of forecast lead time. The medians are calculated over all the runoff realizations for the high-flow cases' set. Both curves are obtained with a moving average method with a temporal window of 5 h. The spread ($q_{90} - q_{10}$) was considered instead of the total spread in order to filter out the ensemble members producing the most extreme values.

### 4.3. Limitations

It is clear that the major limitation of the present study was the limited amount of simulations considered. For a full assessment, this is clearly not enough. However, for a "fit-for purpose" study, i.e., to demonstrate the new method's characteristics and potential, it is sufficient. For this reason, a much wider application of PBM to different periods of the year, different catchments, and with different flood forecasting systems is recommended to check its reliability in different settings, to identify possible defects, and to further improve it. Furthermore, a decisive improvement of the algorithm would be to make it fully objective, without the influence of any arbitrary choice the end-user should make. In fact, the definition of the prominence parameter to detect the relevant runoff peaks is based on the best subjective estimations obtained from the few available forecasts. A wider application of PBM could compensate also for this, producing a more general setting of peak prominence.

## 5. Conclusions

Predicting peak-flow events accurately, caused by heavy rainfalls, for the application of flood relief measures, is extremely important. A statistical and visual method to accomplish this task is the "peak-box" approach [20], which turned out to lack reliability when multiple runoff peaks were present within the forecast. For this reason, a new algorithm, able to produce peak forecasts distinguishing among different peak-flow events within the same forecast, was designed and tested in this work. To do so, a flood forecasting system, formed by the high resolution meteorological 21 member ensemble model COSMO-E and the semi-distributed hydrological model PREVAH, was run in the small Verzasca basin (186 km$^2$), Switzerland, during October and November 2018. The ensemble flood forecasts obtained in this way were used to produce peak-flow estimations through the application of both the former peak-box method and the newly developed one. The latter proved to distinguish correctly among multiple events, to provide sharper and more skillful forecasts than the former approach, and to estimate the peak timings slightly better. A wider application of this newly developed method should be carried out adopting different modeling chains, in different basins, and for a much larger sample of model initializations, in order also to improve its robustness and to further increase the objectivity of the criteria used to design it.

**Author Contributions:** Conceptualization, A.G., M.Z., and M.W.R.; methodology, A.G. and M.Z.; software, A.G.; validation, A.G. and M.Z.; investigation, A.G. and M.Z.; resources, M.Z.; data curation, A.G.; formal analysis, A.G.; writing, original draft preparation, A.G.; writing, review and editing, A.G. and M.W.R.; visualization, A.G.; supervision, M.Z. and M.W.R. All authors have read and agreed to the published version of the manuscript.

**Funding:** This research received no external funding.

**Acknowledgments:** The first author would like to acknowledge the Swiss Federal Institute for Forest, Snow and Landscape Research (WSL), Birmensdorf, Switzerland, for the accommodation received during the elaboration of this work. We thank four anonymous reviewers for their comments and suggestions, which helped ameliorate and clarify the original manuscript.

**Conflicts of Interest:** The authors declare no conflict of interest.

## Abbreviations

The following abbreviations are used in this manuscript:

COSMO     COnsortium for Small-scale MOdeling
D-PHASE     Demonstration of Probabilistic Hydrological and Atmospheric Simulation of Flood Events
ECMWF     European Centre for Medium-range Weather Forecasts
EPS     Ensemble prediction system
HEPS     Hydrological ensemble prediction system
IQR     Interquartile range
MAP     Mesoscale Alpine Programme
NWP     Numerical weather prediction
PBC     Peak-box classic
PBM     Peak-box multipeak
PREVAH     Precipitation-runoff-evapotranspiration HRU model
ROC     Receiver operating characteristic

## Appendix A. Quality of the Forecasts

In this Appendix, we summarize the results of the verification of the ensemble runoff forecasts during the full cases' set of model initializations. Forecast verification is the process of assessing the quality of the forecasts [33]. The verification tools we decided to apply are usually adopted for hydrological ensemble forecasts (e.g., [38]) and are described in detail in [33]. It is clear that the main limitation of this analysis was the very small sample size of only slightly more than a month of simulations considered. This is an important issue and must be taken into account since the verification

tools adopted need fairly large datasets to provide significant information on the performance of the forecast [38].

Figure A1 reports the ROCa (Receiver Operating Characteristic area) [39] time evolution for different runoff thresholds. The forecasts performed well at detecting events and rejecting non-events (ROCa values were always quite close to the perfect forecast value of one). Overall, there was a time dependent increase in forecast quality. This was a confirmation of the results obtained in previous studies [20,37], indicating that the modeling chain formed by coupling COSMO-E with PREVAH provided useful runoff forecasts for decision-makers in the Verzasca basin.

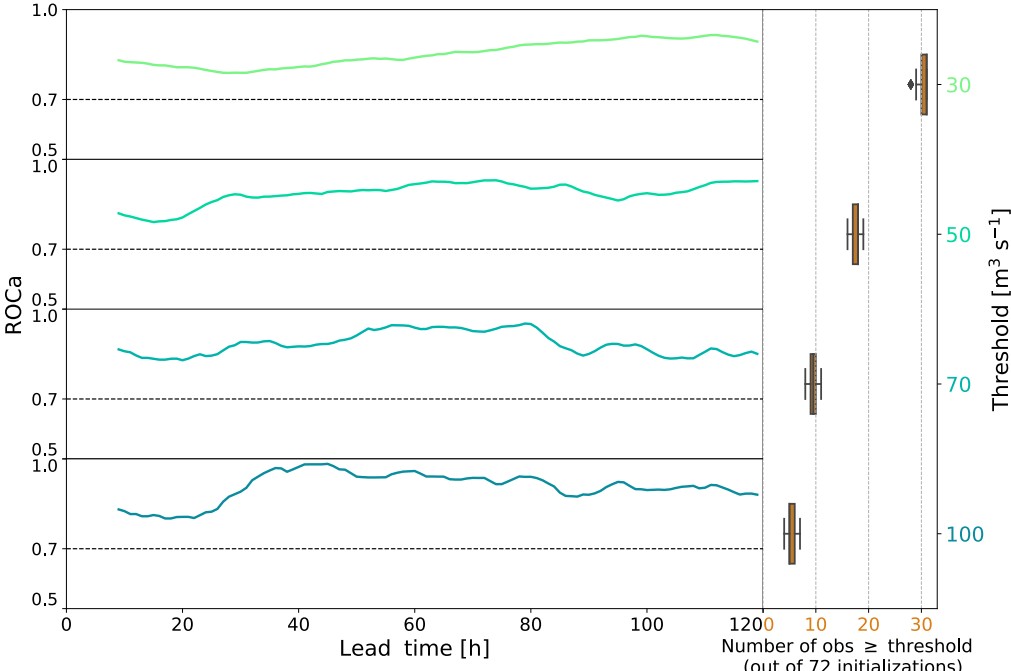

**Figure A1.** Evolution of ROCa with lead time depending on the discharge threshold considering the full cases' set of model initializations. On the right side, the orange boxplots report the numbers of observations exceeding each threshold, which are distributed over the 120 forecast lead times. The black horizontal dashed lines represent the minimum value for which the forecast is still useful for decision makers [40]. The curves are obtained with a moving average method with a 10 h temporal window, applied to smooth out short term fluctuations (this is why the first ten hours of every ROCa time series are missing). In fact, the fluctuations obtained for shorter time periods are another indicator of the limited sample size considered.

Figure A2 reports the reliability diagrams [41] for the full runoff ensemble forecasts and four different thresholds. The forecasts treated were always clearly affected by an unconditional bias causing a persistent underforecasting situation for the Verzasca basin, while conditional biases were not detected. In fact, with the observed relative frequencies always being larger than the respective forecast probabilities, the modeling chain was seen to underpredict the discharge clearly (systematic bias). On the other hand, the slopes of the regression lines (purple solid lines) did not deviate consistently from that of the perfect reliability line (black solid 1:1 line), implying the absence of significant conditional biases. The presence of this negative bias was mainly imputable to a model miscalibration on the particular Verzasca basin itself rather than to the general behavior of the modeling chain adopted, and this was confirmed by the results and comparison of many previous studies [8,9,37,42–44]. This model miscalibration affecting the Verzasca basin has never been fixed after the initial setup performed during the large MAP D-PHASE campaign [45,46], which involved many different catchments in the Alps. The miscalibration has never been corrected because it would have led to a subsequent inability to compare the newest results with all the previous studies. Generally, for these kinds of unconditional

biases, a recalibration of the model on the catchment considered could enhance the reliability of the forecasts and reduce the biases consistently; moreover, also a post-processing step on hydrological ensemble forecasts could be the right way to obtain significantly improved results [47]. The worst forecast reliability estimation was obtained for those events exceeding the lowest runoff threshold. This was imputable to the specific calibration considered for the hydrological model. In fact, we used a calibration expected to give accurate runoff estimations during the peak phases [31], performing consequently worse for levels of discharge near base flow conditions.

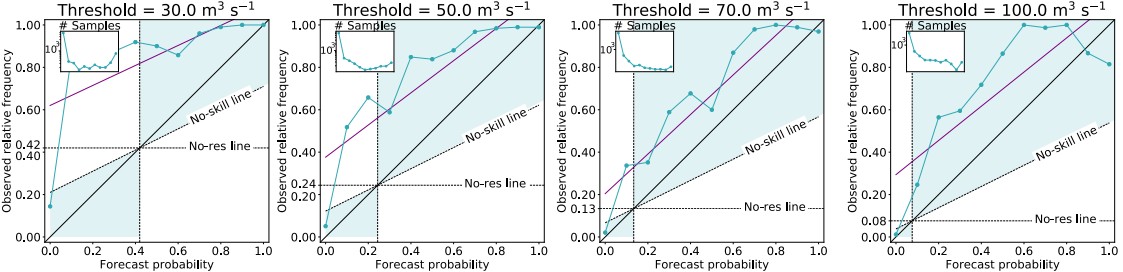

**Figure A2.** Reliability diagrams of runoff ensemble forecasts for various discharge thresholds, eleven different forecast probability classes, and all lead times aggregated (i.e., 120 forecast hours). On the *x*-axis, the forecast probability is reported; on the *y*-axis, the observed relative frequency. The solid black diagonal line indicates the perfect reliability condition; the no-skill line and the no-resolution (no-res) line (i.e., observed climatology value) are also indicated with black dashed lines. The purple solid lines represent the linear regression obtained from the observed relative frequency distribution. The sea-green area represents the portion of the plot for which forecasts contribute positively to the skill. The subplots on the upper left of each panel report the refinement distribution, i.e., the relative frequency of events in the classes.

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
