# Peer review of "Estimating Ensemble Flood Forecasts’ Uncertainty: A Novel “Peak-Box” Approach for Detecting Multiple Peak-Flow Events"

_atmosphere, doi:10.3390/atmos11010002_

Round 1
Reviewer 1 Report
The manuscript is better focused now and I suggest accept.
Reviewer 2 Report
The Authors have replied to my comments appropriately.
The article has gained in clarity by focusing on the PBM.
Long life to the peak-box tool!
Reviewer 3 Report
None
Reviewer 4 Report
The redrafted manuscript focuses on the developments to peak-box algorithm. The aims are now clearly stated, and the overall presentation is much improved. I recommend for publication.
This manuscript is a resubmission of an earlier submission. The following is a list of the peer review reports and author responses from that submission.
Round 1
Reviewer 1 Report
The submitted paper by Giordani et al. discusses two issues: uncertainty sources (including atmospheric forecasts) in probabilistic hydrologic forecasts and a new method for detection of multiple peak-flow events. The latter is highlighted in the title. The former is the issue mentioned first in the abstract. The Conclusion consists of two paragraphs with one for each issue. This is already the main issue I have with the paper. Its goal is not well defined, and it tries too much. Still, the paper’s content is very interesting and in general well written and illustrated.
The paper is submitted to Atmospheres. Given the journal the hydrologic peak-flow method is of less interest for the reader than the impact of atmospheric forecasts on peak-flow forecasting uncertainty. Therefore, I suggest that the authors consider focusing on the hydrometeorological forecasting and moving Sec. 2.6 & 2.7 and some parts of the Results into an appendix. I understand that this would imply changing title, abstract etc. too and implies additional work in the uncertainty part.
A major result is the finding that the meteorological forcing is responsible for an uncertainty 5 times the hydrological uncertainty using the COSMO-E (2.2km) ensemble. Former studies cited (Zappa et al. 2011) imply that COSMO-LEPS (10km) is responsible for 10 times. The authors argue the atmospheric forcing is better because of better COSMO model grid resolution. This asks for a more in-depth discussion: (1) Zappa et al 2011 probably investigated a different period in different catchments(?), (2) What kind of uncertainty is meant in the Conclusion, measured by which skill score? (3) Why grid spacing and not ensemble set-up, other model improvements, or better forcing of the COSMO LAM? Ahrens et al (2003, 10.5194/angeo-21-627-2003) concluded that a coarse model with 10 km grid-spacing provided better input into a hydrological model in the same catchment than the same model with 4km grid-spacing. (5) Jaun et al. (2008, www.nat-hazards-earth-syst-sci.net/8/281/2008/) have shown that COSMO-LEPS using different ensemble sizes provides different peak-spreads doing forecasts with PREVAH. COSMO- & -LEPS have different ensemble sizes. (6) The number of investigated peaks is small (one month). How robust is the conclusion? The 5-day realizations (Sec. 2.4) are not independent. This leads to an uncertainty underestimation?
A few minor issues:
Line 93: usually identified: how done here?
Line 127: “mean discharge” might help the reader
L134: typo: measuerements
L150: “Superposition” implies linearity. I would avoid calling the chaining of errors superposition given the nonlinearities in the hydrological model.
L154: “extremely” in what sense?
L502: “this is must”
Author Response
Authors replies to Reviewer 1:
We want to thank the reviewer for his/her assessment of our manuscript. In the following we give our answers to the comments and recommendations that have been raised. Reviewer comments RC are bold, our reply AR is in italic.
RC: The submitted paper by Giordani et al. discusses two issues: uncertainty sources (including atmospheric forecasts) in probabilistic hydrologic forecasts and a new method for detection of multiple peak-flow events. The latter is highlighted in the title. The former is the issue mentioned first in the abstract. The Conclusion consists of two paragraphs with one for each issue. This is already the main issue I have with the paper. Its goal is not well defined, and it tries too much. Still, the paper’s content is very interesting and in general well written and illustrated.
AR: We thank the reviewer for the remark. We are aware that the paper in this form is too disjointed into two different parts, and this issue has been raised by many of the reviewers involved. For this reason, and given also the very limited sample size that we had available to carry out the simulations for the assessment of different sources of uncertainty and to verify the forecast results, we decided: to remove the part related to the investigation of different uncertainty components and to just briefly discuss it in the Introduction, to consistently reduce and put into an appendix the assessment of the quality of the forecasts, and to focus mainly on the peak-box approach.
RC: The paper is submitted to Atmospheres. Given the journal the hydrologic peak-flow method is of less interest for the reader than the impact of atmospheric forecasts on peak-flow forecasting uncertainty.
AR: Given the name of Atmosphere’s special issue: Climatological and Hydrological Processes in Mountain Regions, we think that this paper fits adequately to the purposes of the journal.
RC: Therefore, I suggest that the authors consider focusing on the hydrometeorological forecasting and moving Sec. 2.6 & 2.7 and some parts of the Results into an appendix. I understand that this would imply changing title, abstract etc. too and implies additional work in the uncertainty part.
AR: We will remove and reduce as explained in the reply to the general comment above, and the different sections involved will be modified adequately.
RC: A major result is the finding that the meteorological forcing is responsible for an uncertainty 5 times the hydrological uncertainty using the COSMO-E (2.2km) ensemble. Former studies cited (Zappa et al. 2011) imply that COSMO-LEPS (10km) is responsible for 10 times. The authors argue the atmospheric forcing is better because of better COSMO model grid resolution. This asks for a more in-depth discussion: (1) Zappa et al 2011 probably investigated a different period in different catchments(?) (2) What kind of uncertainty is meant in the Conclusion, measured by which skill score? (3) Why grid spacing and not ensemble set-up, other model improvements, or better forcing of the COSMO LAM? Ahrens et al (2003, 10.5194/angeo-21-627-2003) concluded that a coarse model with 10 km grid-spacing provided better input into a hydrological model in the same catchment than the same model with 4km grid-spacing. (5) Jaun et al. (2008, www.nat-hazards-earth-syst-sci.net/8/281/2008/) have shown that COSMO-LEPS using different ensemble sizes provides different peak-spreads doing forecasts with PREVAH. COSMO- & -LEPS have different ensemble sizes. (6) The number of investigated peaks is small (one month). How robust is the conclusion? The 5-day realizations (Sec. 2.4) are not independent. This leads to an uncertainty underestimation?
AR: We thank the reviewer for the discussion points presented, but unfortunately we decided to remove the part related to this discussion as explained in the reply to the general comments. Anyway, coming to the questions presented: (1) in Zappa et al. (2011) the same basin investigated here (Verzasca) is treated, together with other catchments, for different periods (they investigated specific events instead of continuous timeseries like the present manuscript), (3) Ahrens et al. (2003) used a deterministic meteorological model (ALADIN), which shall be treated differently from probabilistic ones, especially when comparing it with a case like this where we transit from the convection parametrizing regime (COSMO-LEPS, 10km) to the convection permitting one (COSMO-E, 2.2km), furthermore also in that study a very small sample size (only 3 precipitation events) limits its general validity; (5) concerning the difference in ensemble size, the last part of the thesis from which this paper is extracted (Giordani et al. 2019, University of Innsbruck, ACINN) is related to the ensemble reduction of the meteorological forcing through cluster analysis, and we showed there that with just a third of the members of the full ensemble (7 vs 21), more than the 90% of the spread of the original simulation is covered on average, and that the quality of the forecasts is never substantially decreased (even if also here the limitation is given by the small sample size), confirming previous results found in Verbunt et al. (2007) (doi.org/10.1175/JHM594.1)
RC: A few minor issues:
Line 93: usually identified: how done here?
AR:We will remove the “usually” as the 100 m bands criterion is valid for our case, thank you for the remark.
RC: Line 127: “mean discharge” might help the reader
AR:We will add it, thank you.
L134: typo: measuerements
AR: Thank you for noticing!
L150: “Superposition” implies linearity. I would avoid calling the chaining of errors superposition given the nonlinearities in the hydrological model.
AR: We will remove this part, but thank you for the comment.
L154: “extremely” in what sense?
AR: differing very much from each other, and so producing very different results (i.e. from no precipitation to hundreds of mm/h), anyway also this part will be removed.
L502: “this is must”
AR: Thank you for noticing, we will remove “is”.
Reviewer 2 Report
General comments
Excellent work and interesting paper. The authors succeed in the two open questions they address: about the impact of increased NWP resolution on flood forecast uncertainty and the extension of the “peak-box” approach to the detection of multiple flood peaks.
I found not much to criticize – on the contrary. There is Figure 7 (d) which is a duplicate of (c) instead of illustrating the case of just one peak in the forecast. Apart from this obvious mistake, I have two comments. The first is related to the under-forecasting imputed to a model miscalibration on this particular basin. The reason for this is well explained and the reason for not having corrected it too. It is mentioned on line 525 “a recalibration of the model on the catchment considered could enhance the reliability of the forecasts and reduce consistently the biases”. The reference [45] is on the post-processing of COSMO-LEPS precipitation forecasts using reforecasts. Do the authors mean using post-processed forcing precipitation or post-processing hydrological ensemble forecasts using hydrological reforecasts or even recalibration of the model PREVAH? I would vote for the second. All along the reading the benefit of a post-processing step in the forecasting chain appears worth being tested.
The second comment is about the very small sample size. This limitation doesn’t affect the value of the results at all. The warning is made just a bit late in the text on line 526. This could be reflected in some assertions like those on lines 16 and 600.
Specific comments
L135 measurements
L166 “in simulation mode” instead of “in retrospective mode”. I suppose this mode corresponds to the run of the model up to the initial conditions for the forecasting.
L372 I don’t understand well “the slopes of the reliability lines are not systematically different from the regression through the observations, thus suggesting that the modeling chain is not affected by consistent conditional biases.”
L398 … to four different model initialization times …
Figure 7 Panel (d) is exactly the same as panel (c). Missing this important case. The caption of this figure is excellent but the legend is somewhat too small.
Figure 8. I’m aware of the log scale of Y-axes. However, the horizontal greyish dashed mean PBC line for PB_IQR in the lower panel seems different than the other 3. There are only three grey diamonds lying above this line. Is this mean correct? It would be appropriate to draw confidence intervals for the mean values.
Figure 9. This figure is like a one-band man! Are the empty red and blue boxes in the upper part of the lower plot ordered from top to bottom with increasing lead-time of occurrence?
L499 I don’t understand “This suggests a smoothing operated by the hydrological model ...”. I understand that smoothing is operating but not why the underforecasting bias affecting the entire modeling chain is not simply suggesting a weakness of the model.
L525-531 See general comments.
L585-590 The phrasing could be improved.
Author Response
Authors replies to Reviewer 2:
We want to thank the reviewer for his/her assessment of our manuscript. In the following we give our answers to the comments and recommendations that have been raised. Reviewer comments RC are bold, our reply AR is in italic.
RC: Excellent work and interesting paper. The authors succeed in the two open questions they address: about the impact of increased NWP resolution on flood forecast uncertainty and the extension of the “peak-box” approach to the detection of multiple flood peaks.
I found not much to criticize – on the contrary. There is Figure 7 (d) which is a duplicate of (c) instead of illustrating the case of just one peak in the forecast. Apart from this obvious mistake, I have two comments. The first is related to the under-forecasting imputed to a model miscalibration on this particular basin. The reason for this is well explained and the reason for not having corrected it too. It is mentioned on line 525 “a recalibration of the model on the catchment considered could enhance the reliability of the forecasts and reduce consistently the biases”. The reference [45] is on the post-processing of COSMO-LEPS precipitation forecasts using reforecasts. Do the authors mean using post-processed forcing precipitation or post-processing hydrological ensemble forecasts using hydrological reforecasts or even recalibration of the model PREVAH? I would vote for the second. All along the reading the benefit of a post-processing step in the forecasting chain appears worth being tested.
The second comment is about the very small sample size. This limitation doesn’t affect the value of the results at all. The warning is made just a bit late in the text on line 526. This could be reflected in some assertions like those on lines 16 and 600.
AR: We thank the reviewer for the comments. We believe that the paper in this form is too much disjointed into two different parts, and this issue has been raised by many of the reviewers involved. Furthermore, while the limitation of the sample size does not constitute a problem when coming to the evaluation of the new peak-box method and its comparison with the former approach, as you also pointed out, it becomes relevant for the study of different sources of uncertainty and for the evaluation of the quality of the forecasts, and this was the main issue the other reviewers criticized about the manuscript. For these reasons we decided: to remove the part related to the investigation of different uncertainty components and to just briefly discuss it in the Introduction, to consistently reduce and put into an appendix the assessment of the quality of the forecasts, and to focus mainly on the peak-box approach.
Concerning the reference [45], we think that a recalibration of the model PREVAH on this basin is necessary in order to obtain results not affected by strong unconditional biases, but, as you also suggested, it would be also interesting to investigate whether the usage of post-processed hydrological simulations could improve the results. We will make it present in the appendix when presenting the assessment of the quality of the forecasts, thank you for the suggestion!
Concerning the warning on the small sample size, we will adjust to make it clear from the beginning.
Specific comments
RC: L135 measurements
AR: Thank you for noticing!
RC: L166 “in simulation mode” instead of “in retrospective mode”. I suppose this mode corresponds to the run of the model up to the initial conditions for the forecasting.
AR: By “retrospective” we intend that the meteorological forcing to the hydrological model is given by the observed precipitation and not by simulations, so that the different ensemble members are related only to different choices of hydrological parameters, and not to different meteorological initial conditions. This part will be removed, but thank you for the comment.
RC: L372 I don’t understand well “the slopes of the reliability lines are not systematically different from the regression through the observations, thus suggesting that the modeling chain is not affected by consistent conditional biases.”
AR: Actually you are right, this sentence is not very clear, thank you for noticing. We will change it to: “the slopes of the regression lines (purple solid lines) do not deviate consistently from that of the perfect-reliability line (black solid 1:1 line), implying the absence of significant conditional biases. ”
RC: L398 … to four different model initialization times …
AR: We will adjust, thank you.
RC: Figure 7 Panel (d) is exactly the same as panel (c). Missing this important case. The caption of this figure is excellent but the legend is somewhat too small.
AR: Thank you very much for noticing! And sorry for the mistake. We will change with the correct figure, and we will also try to enlarge the legend.
RC: Figure 8. I’m aware of the log scale of Y-axes. However, the horizontal greyish dashed mean PBC line for PB_IQR in the lower panel seems different than the other 3. There are only three grey diamonds lying above this line. Is this mean correct? It would be appropriate to draw confidence intervals for the mean values.
AR: Thank you very much for this comment!! We checked and there was an error on the plotting code: the mean is actually the correct one, but the data points were not the last updated ones. We will update with the correct figure.
RC: Figure 9. This figure is like a one-band man! Are the empty red and blue boxes in the upper part of the lower plot ordered from top to bottom with increasing lead-time of occurrence?
AR: Thank you for the comment! The only three cases for which we have both empty red and blue boxes, for which it is actually relevant to know if the empty boxes are ordered with increasing leadtime of occurrence, correspond to the cases in which PBM reduces to PBC, and both miss the observation, so the empty boxes of the two methods result to be exactly equal, both in terms of sharpness, median and leadtime of occurrence. We will add a comment about it on figure’s caption.
RC: L499 I don’t understand “This suggests a smoothing operated by the hydrological model ...”. I understand that smoothing is operating but not why the underforecasting bias affecting the entire modeling chain is not simply suggesting a weakness of the model.
AR: This part will be removed.
RC: L525-531 See general comments.
AR: See the reply to the general comments.
RC: L585-590 The phrasing could be improved.
AR: That is right, anyway we will substantially revise the conclusions removing that part.
Reviewer 3 Report
This paper announces improvements to the "Peak-box" approach for detecting multiple peak-flow events. Peak-boxes are quite useful for interpreting uncertainties in the timing and in the intensity of the occurrence of a coming flood, as described by an ensemble. This is a much welcomed proposition as the peak-box methodology does not intrinsically account for multiple peaks.
I am overall quite pleased by this paper. My main problem is that the introduction announces two objectives: evaluation of a novel COSMO H-EPS implementation and development a peak-box approach for detecting multiple flood peaks in ensemble forecasts. I wished that the paper would have been limited to only the second objective, as announced by the title of the manuscript. I thus strongly recommend that the manuscript be revised eliminating parts that deal with COSMO verification: sections 2.7, 3.2 and 4.2, and Figures 4, 5 and 6. The conclusion will have to be shorten as well. Those elements are not needed to explain and interpret the peak-box methodology and results. Furthermore, since the flood forecasting system was run from 2018-10-23 12:00 UTC to 2018-11-28 12:00 UTC (line 117), it forms a quite short experiment to assess its quality. The series are, however, suitable to test a novel Peak-box methodology (line 570).
Other comments
Line 41: you may consider as well:
Boelee L, Lumbroso DM, Samuels PG, Cloke HL. 2018. Estimation of uncertainty in flood forecasts – A comparison of methods. Journal of Flood Risk Management, doi.org/10.1111/jfr3.12516. Thiboult A, Anctil F, Boucher MA. 2016. Accounting for three sources of uncertainty in ensemble hydrological forecasting. Hydrology and Earth System Sciences 20, 1809-1825.
Line 43: “the uncertainty arising from the usage of different hydrological model parameters due to incomplete process representation”. I would rephrase that. Parameter uncertainty is typically much lesser than the model uncertainty.
Line 102: More details should be given, such as objective function, optimization tool, and validation period. The calibration period is given below.
Line 128: If the threshold is exceeded only 10 % of the time, it is in practice quite high, not low. Please clarify.
Line 131: Please specify how the interpolation was performed.
Figure 2: For consistency, the hydrograph is the first two panels should also be present in the last one. An alternative would be to create two distinct figures, isolating the last panel.
Line 207 (and other similar): “A critical limitation of PBC is its inability to forecast peak flows”. I disagree with the usage of “forecast”. I would prefer “interpret”, as the forecast is performed by the H-EPS.
Line 303: “21 meteorological medians out of the 525 model predictions”. Is this a general recommendation?
Figure 7: It seems that panel d is just the repetition of panel c. It does not correspond to the text...
Author Response
Authors replies to Reviewer 3:
We want to thank the reviewer for his/her assessment of our manuscript. In the following we give our answers to the comments and recommendations that have been raised. Reviewer comments RC are bold, our reply AR is in italic.
RC: This paper announces improvements to the "Peak-box" approach for detecting multiple peak-flow events. Peak-boxes are quite useful for interpreting uncertainties in the timing and in the intensity of the occurrence of a coming flood, as described by an ensemble. This is a much welcomed proposition as the peak-box methodology does not intrinsically account for multiple peaks.
I am overall quite pleased by this paper. My main problem is that the introduction announces two objectives: evaluation of a novel COSMO H-EPS implementation and development a peak-box approach for detecting multiple flood peaks in ensemble forecasts. I wished that the paper would have been limited to only the second objective, as announced by the title of the manuscript. I thus strongly recommend that the manuscript be revised eliminating parts that deal with COSMO verification: sections 2.7, 3.2 and 4.2, and Figures 4, 5 and 6. The conclusion will have to be shorten as well. Those elements are not needed to explain and interpret the peak-box methodology and results. Furthermore, since the flood forecasting system was run from 2018-10-23 12:00 UTC to 2018-11-28 12:00 UTC (line 117), it forms a quite short experiment to assess its quality. The series are, however, suitable to test a novel Peak-box methodology (line 570).
AR: We thank the reviewer for the comments. We agree with you: we admit that the paper in this form is too much disjointed into two different parts, and this issue has been raised also by other reviewers involved. We also acknowledge the limitation to the assessment of the quality of the forecasts for such a small sample size. For these reasons we decided to do as you suggested: to remove the part related to the investigation of different uncertainty components and to just briefly discuss it in the Introduction, to consistently reduce and put into an appendix the assessment of the quality of the forecasts (and not to eliminate it in its entirety because we believe that some evaluation of the quality of the forecasts should be attached in order to give the right weight to the peak-box results), and to focus mainly on the peak-box approach.
RC: Other comments
Line 41: you may consider as well:
Boelee L, Lumbroso DM, Samuels PG, Cloke HL. 2018. Estimation of uncertainty in flood forecasts – A comparison of methods. Journal of Flood Risk Management, doi.org/10.1111/jfr3.12516. Thiboult A, Anctil F, Boucher MA. 2016. Accounting for three sources of uncertainty in ensemble hydrological forecasting. Hydrology and Earth System Sciences 20, 1809-1825.
AR: Thank you for the suggestion, we will add the references.
RC: Line 43: “the uncertainty arising from the usage of different hydrological model parameters due to incomplete process representation”. I would rephrase that. Parameter uncertainty is typically much lesser than the model uncertainty.
AR: We will remove this part.
RC: Line 102: More details should be given, such as objective function, optimization tool, and validation period. The calibration period is given below.
AR: We will add some details, anyway the reader should be referred to the references for a complete description of the calibration considered.
RC: Line 128: If the threshold is exceeded only 10 % of the time, it is in practice quite high, not low. Please clarify.
AR: For low flow we intend, for this particular basin, those usual river conditions that are largely exceeded when peak flow events happen, which can exceed hundreds of m3s-1. So “low“ in this case is an objective adjective.
RC: Line 131: Please specify how the interpolation was performed.
AR: We will add “The interpolation of precipitation gauges is done with the inverse distance weighting method (the reader is referred to Viviroli et al. (2009) for further details).”
RC: Figure 2: For consistency, the hydrograph is the first two panels should also be present in the last one. An alternative would be to create two distinct figures, isolating the last panel.
AR: The single runoff simulation present in the first two panels is actually present also in the third one, it may not so visible at a glance due to the different runoff range covered by the vertical axis.
RC: Line 207 (and other similar): “A critical limitation of PBC is its inability to forecast peak flows”. I disagree with the usage of “forecast”. I would prefer “interpret”, as the forecast is performed by the H-EPS.
AR: That is a good point, we will change accordingly, thank you.
RC: Line 303: “21 meteorological medians out of the 525 model predictions”. Is this a general recommendation?
AR: For the cases we treated we noticed that considering just the meteorological medians instead of the entire set of ensemble simulations with varied hydrological parameters, the peak detection procedure was not significantly affected, especially concerning the timing of the peak which is the most important information to achieve. We think that for a comparison study like this, this is a valid criterion, but for other circumstances further investigation shall be recommended. We will add a comment about that.
RC: Figure 7: It seems that panel d is just the repetition of panel c. It does not correspond to the text…
AR: Thank you for noticing! We will update the figure with the correct panel.
Reviewer 4 Report
The article presents results for a 1 month case-study period of the semi-distributed hydrological catchment model PREVAH, with inputs from the COSMO-E high resolution precipitation ensemble prediction system. The uncertainties of the forecasting system are examined comparing the precipitation uncertainty to that from a multi-physics hydrological ensemble. Additionally, the Peak-box approach of Zappa et al. 2012 is further developed to account for multiple peaks in a forecast. These are important, and operationally relevant topics of investigation, and I fully support their publication. However, there are several aspects of the presentation and organisation of the article which I feel require significant editing, and potentially further investigations.
General comments
The main aims of the article are unclear, and vary throughout. For example, the abstract states “This paper aims at studying uncertainties”, but later in section 1.1 the main questions are around model resolution and the peak-box approach development. The aims need to be clarified and consistently presented. The results section appears similarly disjointed – first looking at existing metrics to assess uncertainties, then secondly and separately developing the peak-box approach. Although I understand the authors’ position that using the peak-box approach to assess the uncertainties would require large computational resources, I consider this comparison essential to demonstrate this new and valuable method. As a minimum, further discussion should be added comparing the peak-box results with the traditional metrics in the context of understanding the forecast uncertainties. Section 3.3 currently focuses on comparing instead the new and old peak-box methods. The results section, and associated discussion, need to be revisited with sampling uncertainty considered and discussed throughout. Although the sampling issue is highlighted at the end of section 4.2 and in 4.3.2, it is critical for the robust interpretation of results and should be discussed earlier. I also recommend that uncertainty estimates are included in figures 4, 5 and 6, for example using a bootstrap approach. This would help clarify the statistical relevance of small differences in performance. In figure 4 is it meaningful to show the higher threshold results, or is this misleading information? Throughout the article the language should be formalised, avoiding phrases/words such as “basically”, “a couple of times”, “turned out to be”, “attempt” and “not so much”, and writing out small numbers in text. Please also avoid the use of # to mean number in text and equations.Specific comments
Consider removing/rewording L9-11, given that the forcing with a lower resolution model is discussed for context and comparison, and was done in a previous study. Please back this up with references, or be more specific, or both. I presume the authors mean to say “understanding of probabilistic runoff forecasts”? I presume the authors mean 500m2 The reader is referred to … Section 1.1. Consider re-writing as discussed in general comments above. Section 2.2. Please comment on the representivity of using only one river gauge, and the uncertainty in precipitation estimates due to interpolation of the raingauge data, particularly in mountainous terrain. This is important information to put the results in context. Please back up this statement with references. It will never be possible to account for the total uncertainty in the modelling chain, as there will always be model limitations and “unknown unknowns”. Please rephrase. Section 2.7 Opening. In this context I would expect the aim of the verification to be “better understanding the forecast uncertainties, and investigating what insight can be gained from the new peak-box approach”. Please also refer back to the key paper questions here. Please refer to the original verification metric papers in section 2.7 Section 2.7.2, final sentence. Either state how one can understand, or remove this sentence. L217 “for both the proposed approaches”. The first approach has already been published so is not being proposed here. Please rephrase. The discussion of model bias and calibration in Section 4.2 is critical to the interpretation and understanding of the results but comes too late. Consider moving this discussion to the Methods section, and discussing alongside the results in Section 3. Figure 4. Please comment further on the effect of the temporal averaging. Where other averaging periods considered? What is causing the fluctuations in performance at shorter time periods? Would it make more sense to pool the data in time before calculating the verification statistics instead of applying a retrospective temporal average? Figure 5. Please consider the appropriateness of the regression lines given the highly non-linear dependence of the results on forecast probability. For example, “Threshold 100m3s-1“ shows a totally different trend above a forecast probability of 0.6. The title of Section 2.2 needs clarifying. Is this performance of the new PBM approach, or performance of the models evaluated using this approach? L570-572. The number of simulations is a limitation for all parts of this study, not just the assessment of the peak-box approach. Please expand. Please rephrase. Although the peak-box is a useful tool, many other things are also needed to predict peak-flow events in a useful and meaningful manner.
Author Response
Authors replies to Reviewer 4:
We want to thank the reviewer for his/her assessment of our manuscript. In the following we give our answers to the comments and recommendations that have been raised. Reviewer comments RC are bold, our reply AR is in italic.
RC: The article presents results for a 1 month case-study period of the semi-distributed hydrological catchment model PREVAH, with inputs from the COSMO-E high resolution precipitation ensemble prediction system. The uncertainties of the forecasting system are examined comparing the precipitation uncertainty to that from a multi-physics hydrological ensemble. Additionally, the Peak-box approach of Zappa et al. 2012 is further developed to account for multiple peaks in a forecast. These are important, and operationally relevant topics of investigation, and I fully support their publication. However, there are several aspects of the presentation and organisation of the article which I feel require significant editing, and potentially further investigations.
General comments
The main aims of the article are unclear, and vary throughout. For example, the abstract states “This paper aims at studying uncertainties”, but later in section 1.1 the main questions are around model resolution and the peak-box approach development. The aims need to be clarified and consistently presented. The results section appears similarly disjointed – first looking at existing metrics to assess uncertainties, then secondly and separately developing the peak-box approach. Although I understand the authors’ position that using the peak-box approach to assess the uncertainties would require large computational resources, I consider this comparison essential to demonstrate this new and valuable method. As a minimum, further discussion should be added comparing the peak-box results with the traditional metrics in the context of understanding the forecast uncertainties. Section 3.3 currently focuses on comparing instead the new and old peak-box methods. The results section, and associated discussion, need to be revisited with sampling uncertainty considered and discussed throughout. Although the sampling issue is highlighted at the end of section 4.2 and in 4.3.2, it is critical for the robust interpretation of results and should be discussed earlier. I also recommend that uncertainty estimates are included in figures 4, 5 and 6, for example using a bootstrap approach. This would help clarify the statistical relevance of small differences in performance. In figure 4 is it meaningful to show the higher threshold results, or is this misleading information? Throughout the article the language should be formalised, avoiding phrases/words such as “basically”, “a couple of times”, “turned out to be”, “attempt” and “not so much”, and writing out small numbers in text. Please also avoid the use of # to mean number in text and equations.
AR: We thank the reviewer for the comments. We agree with the fact that the paper in this form is too much disjointed into two different parts, and this issue has been raised also by other reviewers involved. Furthermore, while the limitation of the sample size does not constitute a problem when coming to the evaluation of the new peak-box method and its comparison with the former approach, it becomes relevant for the study of the different sources of uncertainty and for the evaluation of the quality of the forecasts, and this was the main issue other reviewers also criticized about the manuscript. For these reasons we decided: to remove the part related to the investigation of different uncertainty components and to just briefly discuss it in the Introduction, to consistently reduce and put into an appendix the assessment of the quality of the forecasts, and to focus mainly on the peak-box approach. Concerning the latter, we believe that the comparison with the former peak-box method is the best way to assess the added value presented by the newly developed one, since the main focus of this work is to ameliorate the original method in those occasions for which it is not able to produce reliable results. Consequently, the comparison of the new method with other metrics, despite it would be certainly extremely valuable, is however beyond the scope of this manuscript. Concerning the warning on the small sample size, we will adjust to make it clear from the beginning. We will also remove the highest threshold from the reliability and ROCa diagrams, because, as you noted, the batch of data from which they are calculated is not enough to provide useful information. We will also formalize the language and remove the # characters.
Specific comments
RC: Consider removing/rewording L9-11, given that the forcing with a lower resolution model is discussed for context and comparison, and was done in a previous study. Please back this up with references, or be more specific, or both.
AR: We will remove this part.
RC: I presume the authors mean to say “understanding of probabilistic runoff forecasts”?
AR: Yes, thank you for the comment, we will revise accordingly.
RC: I presume the authors mean 500m2
AR: Yes, thank you for noticing, we will correct.
RC: The reader is referred to …
AR: We will correct, thank you.
RC: Section 1.1. Consider re-writing as discussed in general comments above.
AR: We will revise accordingly to the reply to the general comments.
RC: Section 2.2. Please comment on the representivity of using only one river gauge, and the uncertainty in precipitation estimates due to interpolation of the raingauge data, particularly in mountainous terrain. This is important information to put the results in context. Please back up this statement with references.
AR: We do not think that using only one river gauge constitute a problem for our case, especially because this manuscript does not want to find and discuss results that are valid generally for every type of catchments and in every conditions, it is thought to be more of a technical paper for specific basins, like the one treated. It is clear that for basins with different nature (e.g. comprising a lake) the peak-box approach would not be needed, while for basins like Verzasca where in e.g. 3-4 days in a row multiple peak-flow events can take place, the peak-box could be helpful. Concerning the uncertainty rising from the interpolation of raingauge data, despite the important impact it can have, as you highlighted, we believe that this goes beyond the scope of the paper, since we are not focusing on forecasting and verify precipitation estimates, but our focus is on the comparison between the former and the new peak-box methods, and the improvements obtained with the latter.
RC: It will never be possible to account for the total uncertainty in the modelling chain, as there will always be model limitations and “unknown unknowns”. Please rephrase.
AR: That is absolutely right, thank you for the comment, we will rephrase accordingly.
RC: Section 2.7 Opening. In this context I would expect the aim of the verification to be “better understanding the forecast uncertainties, and investigating what insight can be gained from the new peak-box approach”. Please also refer back to the key paper questions here.
AR: We will rephrase and add the relevant references, thank you.
RC: Please refer to the original verification metric papers in section 2.7
AR: We will add the relevant references, thank you.
RC: Section 2.7.2, final sentence. Either state how one can understand, or remove this sentence.
AR: This subsection will be removed.
RC: L217 “for both the proposed approaches”. The first approach has already been published so is not being proposed here. Please rephrase.
AR: Thank you for noticing, we will rephrase accordingly.
RC: The discussion of model bias and calibration in Section 4.2 is critical to the interpretation and understanding of the results but comes too late. Consider moving this discussion to the Methods section, and discussing alongside the results in Section 3.
AR: Since we re-organize this entire part, we will move this discussion earlier as you suggested, thank you.
RC: Figure 4. Please comment further on the effect of the temporal averaging. Where other averaging periods considered? What is causing the fluctuations in performance at shorter time periods? Would it make more sense to pool the data in time before calculating the verification statistics instead of applying a retrospective temporal average?
AR: We will expand the comment on temporal averaging, anyway this part will be consistently reduced as noted in the reply to the general comments above.
RC: Figure 5. Please consider the appropriateness of the regression lines given the highly non-linear dependence of the results on forecast probability. For example, “Threshold 100m3s-1“ shows a totally different trend above a forecast probability of 0.6.
AR: This issue is relevant and mainly caused by the limitation of the sample size, anyway we will reduce much of this part.
RC: The title of Section 2.2 needs clarifying. Is this performance of the new PBM approach, or performance of the models evaluated using this approach?
AR: Do you mean Section 3.3? In that case it will be removed as the Results section will be focused only on the peak-box results.
RC: L570-572. The number of simulations is a limitation for all parts of this study, not just the assessment of the peak-box approach. Please expand. Please rephrase. Although the peak-box is a useful tool, many other things are also needed to predict peak-flow events in a useful and meaningful manner.
AR: See above the reply to the general comments.